# Maintained avalanche dynamics during task-induced changes of neuronal activity in nonhuman primates

Shan Yu[1†‡], Tiago L Ribeiro[1†], Christian Meisel[1], Samantha Chou[1], Andrew Mitz[2], Richard Saunders[2], Dietmar Plenz[1]*

[1]Section on Critical Brain Dynamics, National Institute of Mental Health, Bethesda, United States; [2]Laboratory of Neuropsychology, National Institute of Mental Health, Bethesda, United States

*For correspondence: plenzd@mail.nih.gov

[†]These authors contributed equally to this work

Present address: [‡]Institute of Automation, and Center for Excellence in Brain Science and Intelligence Technology, Chinese Academy of Sciences, Beijing, China

Competing interests: The authors declare that no competing interests exist.

**Abstract** Sensory events, cognitive processing and motor actions correlate with transient changes in neuronal activity. In cortex, these transients form widespread spatiotemporal patterns with largely unknown statistical regularities. Here, we show that activity associated with behavioral events carry the signature of scale-invariant spatiotemporal clusters, neuronal avalanches. Using high-density microelectrode arrays in nonhuman primates, we recorded extracellular unit activity and the local field potential (LFP) in premotor and prefrontal cortex during motor and cognitive tasks. Unit activity and negative LFP deflections (nLFP) consistently changed in rate at single electrodes during tasks. Accordingly, nLFP clusters on the array deviated from scale-invariance compared to ongoing activity. Scale-invariance was recovered using 'adaptive binning', that is identifying clusters at temporal resolution given by task-induced changes in nLFP rate. Measures of LFP synchronization confirmed and computer simulations detailed our findings. We suggest optimization principles identified for avalanches during ongoing activity to apply to cortical information processing during behavior.
DOI: https://doi.org/10.7554/eLife.27119.001

## Introduction

Neuronal activity in the brain has been traditionally separated into ongoing activity, which lacks a particular sensory stimulus or movement, and evoked activity, which is the response to a well-defined stimulus. Yet, studies have consistently shown that ongoing activity predicts stimulus response (*Arieli et al., 1996*; *Tsodyks et al., 1999*; *Luczak et al., 2009*) and behavioral outcome (*Supèr et al., 2003*; *Womelsdorf et al., 2006*) suggesting that both forms of activity are closely related. During the last decade, ongoing activity has been found to organize as neuronal avalanches (*Beggs and Plenz, 2003*) — scale-invariant activity cascades whose size, duration, waveform and inter-cascade intervals are governed by power laws (for review see *Chialvo (2010)* and *Plenz (2012)*). From microscale to macroscale organization, neuronal avalanches describe spontaneous firing of local pyramidal neuron groups in awake rodents in vivo (*Bellay et al., 2015*), the ongoing local field potential (LFP) and subthreshold population activity in rodents and nonhuman primates (*Gireesh and Plenz, 2008*; *Petermann et al., 2009*; *Scott et al., 2014*), as well as resting activity in humans observed using functional magnetic resonance imaging (fMRI) (*Fraiman and Chialvo, 2012*; *Tagliazucchi et al., 2012*; *Haimovici et al., 2013*), magnetoencephalography (MEG) (*Palva et al., 2013*; *Shriki et al., 2013*) and electrocorticography (ECoG) (*Priesemann et al., 2014*). Neuronal avalanches signify a cortical network at a critical or near-critical state (*Chialvo, 2010*; *Plenz, 2012*), which experiments (*Shew et al., 2009*; *Gautam et al., 2015*) and simulations (*Beggs and Plenz, 2003*; *Bertschinger and Natschläger, 2004*; *Haldeman and Beggs, 2005*; *Kinouchi and Copelli,*

*2006*; *Rämö et al., 2007*; *Nykter et al., 2008*) have shown to hold numerous advantages in information processing, such as maximum mnemonic repertoire size (*Haldeman and Beggs, 2005*; *Shew et al., 2011*; *Fagerholm et al., 2016*), information diversity (*Nykter et al., 2008*) and dynamic range (*Kinouchi and Copelli, 2006*; *Shew et al., 2009*; *Gautam et al., 2015*), optimized computational capabilities (*Bertschinger and Natschläger, 2004*), information transmission (*Beggs and Plenz, 2003*; *Rämö et al., 2007*; *Fagerholm et al., 2016*), sensory discrimination (*Tomen et al., 2014*) and learning (*de Arcangelis and Herrmann, 2010*). Yet, despite these potential advantages, evidence for neuronal avalanches outside the realm of ongoing activity, specifically, during behaviorally relevant motor and cognitive tasks, has been controversial.

Recent analysis of neuronal avalanches during sustained dynamic movie stimulation in the visual cortex of turtle (*Shew et al., 2015*) and visuo-motor tasks in human fMRI (*Fagerholm et al., 2015*) and EEG (*Arviv et al., 2015*) suggest transient dynamics that differ from scale-invariant avalanches. Similarly, neural models implied transient deviations from avalanche dynamics during strong external inputs (*Millman et al., 2010*; *Taylor et al., 2013*; *Hartley et al., 2014*; *Stepp et al., 2015*; *Williams-García et al., 2014*). Here, we hypothesize that transient changes in activity during evoked responses could simply reflect a change in the rate of avalanches rather than a transition to a different dynamical regime, for example a change in synchronization. Such a rate change likely requires modifications to standard avalanche analysis, as the standard was originally introduced for stationary rates of activity (*Beggs and Plenz, 2003*). In fact, by taking changes in activity rate into account, here we show in the behaving nonhuman primate that cortical dynamics during sensory and cognitive processing exhibit clear power law organization in line with the maintenance of neuronal avalanches. In line with these findings, our phase-synchronization analysis demonstrates that synchrony levels are maintained for ongoing and task-related activity. We expand on our experimental results using computational models and explore the sensitivity of this advanced approach to distinguish avalanche dynamics from states with changed synchrony, for example supercritical dynamics. Our results extend previous work on the dynamics of resting state activity and suggest that the optimization principles identified for critical dynamics during resting state activity extend to and sculpt cortical information processing during behavior.

## Results

In order to demonstrate avalanche maintenance during evoked activity that is independent of specific brain region or behavioral paradigm, we chose two types of tasks and two cortical areas in two adult macaque monkeys. High-density microelectrode arrays (MEA; 10 × 10; corner electrodes missing; inter electrode distance Δd = 400 µm) were chronically implanted in left prefrontal cortex (PFC) of monkey A and left premotor cortex (PM) of monkey B. We recorded extracellular unit activity (0.3–3 kHz) to confirm involvement of recording sites in behavioral tasks as well as the LFP (1–100 Hz), which is commonly used for avalanche analysis (*Beggs and Plenz, 2003*; *Petermann et al., 2009*; *Scott et al., 2014*).

### Avalanche maintenance during a cue-triggered cognitive task

In our first behavioral condition, we asked whether scale-invariant dynamics can be found when strong transient activity changes due to cognitive processing are present. To this end, we analyzed avalanche dynamics in the dorsal-lateral prefrontal cortex (dlPFC; left hemisphere) for a visual-motor mapping task (monkey A; *Figures 1–3*). Monkey A applied a learned rule (>95% success rate) to two individually presented visual cues, which instructed the retrieval of food from a left feeder (left trials) or right feeder (right trials).

It is known that the dlPFC is involved in such visual-motor mapping task (*Asaad et al., 1998*; *Puig and Miller, 2012*). Consistently, in all of 29 putative single-units recorded in the dlPFC (1 day, 1 hr recording session), unit activity was variable with transient increases or decreases in firing rate around cue on- and off-set that depended on left or right trials (*Figure 1A*; three examples shown). No significant change in Fano Factor (*Churchland et al., 2010*) was found during task execution (*Figure 1B*; see Materials and methods). On the other hand, the average LFP changed consistently for each electrode exhibiting two waveforms that differed between left and right trials (*Figure 1C*). We extracted peak times of the negative LFP (nLFP) at a threshold *thr* = –2 SD for further analysis (see Materials and methods), and observed a rapid ~8-fold increase in nLFP rate ~0.5 s after cue

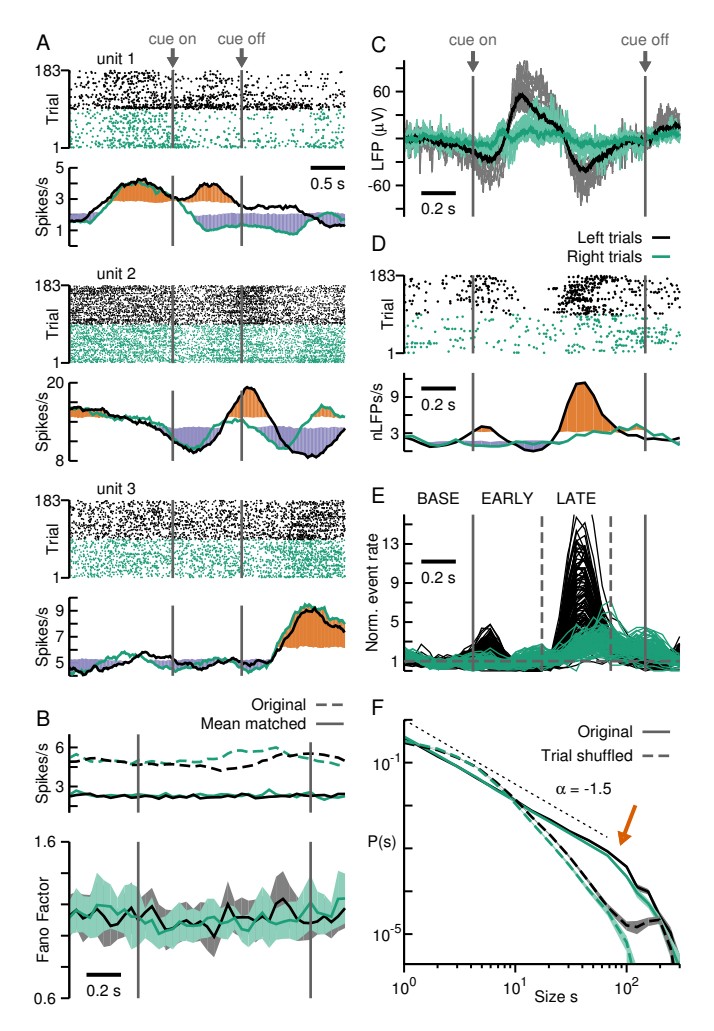

**Figure 1.** Neuronal avalanches in prefrontal cortex during a cognitive task. (A) Firing rate changes for putative single units demonstrate PFC region recorded by the array is involved in task performance. Unit raster (*top*) separated into right (*green*) and left (*black*) trials with corresponding average firing rates (*bottom*). Cue presentation elicits distinct rate changes for units 1 and 2 during left and right trials, but not for unit 3. Colored area (*orange/purple*) indicates periods that differ significantly (high/low) from baseline (see Materials and methods). *Vertical solid lines*: Cue on- and off-set. (B) The population of PFC units does not show a task-related change in rate (*top*) or Fano Factor (*bottom*). *Top*: mean-matched/original rate in *solid/broken* lines; 38/48% survived matching for left/right trials. *Bottom*: mean-matched spikes. *Shaded areas*: 95% confidence interval from linear regression per time window. (C) Distinct changes in the LFP for right (*green*) and left (*black*) trials during the task. *Grey/light green*: trial averages for each electrode. *Black/dark green*: average over electrodes. (D) The negative LFP (nLFP; −2SD threshold; *dots*) allows for distinguishing right (*green*) from left (*black*) trials. *Top*: Example nLFP raster (single electrode) separated into right (*green*) and left (*black*) trials. *Bottom*: Corresponding time course in average nLFP rate. Colored areas (*orange/purple*) indicate significant change (high/low) from baseline. (E) Distinct change in nLFP rate separates a baseline (BASE; 400 ms) period from an early (EARLY; 400 ms) and late (LATE; 400 ms) epoch after cue onset (*vertical lines*). (F) nLFPs on the array when analyzed for full left and right trial periods show avalanche organization. Power law in size probability densities for nLFP clusters (*solid*; fixed Δt). *Arrow*: Cut-off at number of electrodes on the array. *Broken lines*: Corresponding trial-shuffle controls. *Shaded areas*: Corresponding 95% confidence interval based on bootstrapping. *Dotted line*: Visual guide for exponent of −1.5.

DOI: https://doi.org/10.7554/eLife.27119.002

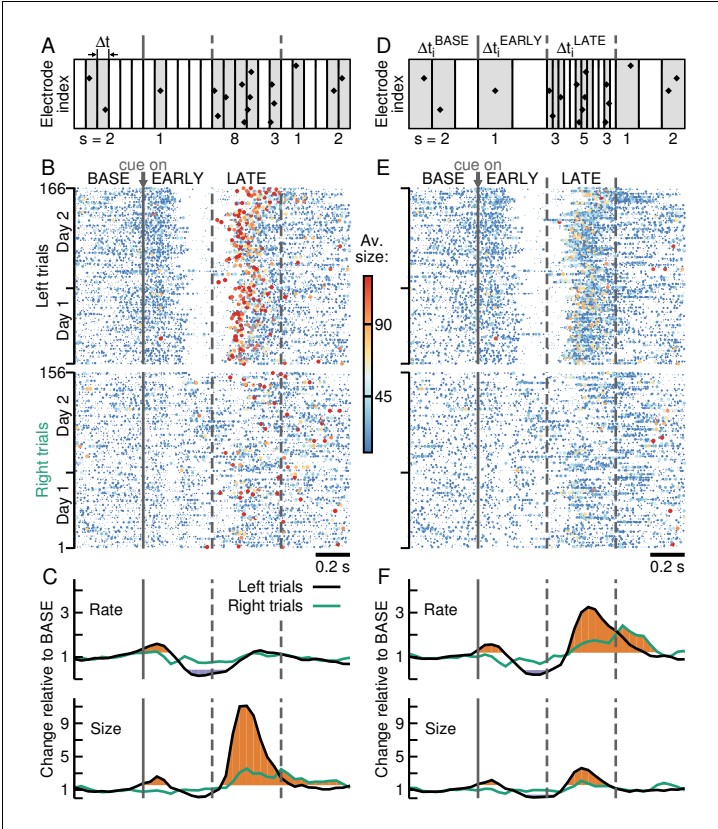

**Figure 2.** Adaptive binning tracks avalanche organization during behaviorally induced transient activity changes in PFC. (**A**) Fixed binning defines clusters (*grey area*) by successive time bins of constant duration Δt with at least one nLFP (*diamonds*). The temporal resolution Δt does not change for trials or epochs. Size *s* is defined as the number of nLFPs per cluster. (**B**) Large avalanches (*red*) dominate for left trials during LATE. Avalanche raster with size coded by color and dot size at fixed Δt for left (*top*) and right (*bottom*) trials for 2 consecutive recording days. (**C**) Significant increases (*orange*) and decreases (*purple*) in average time course for avalanche rate (*top*) and size (*bottom*) for left (*black*) and right (*green*) trials. (**D**) Adaptive binning links Δt for each trial *i* to the average nLFP rate during epochs resulting in three different temporal resolutions for cluster definition: $\Delta t_i^{BASE,EARLY,LATE}$ (cp. **A**). (**E**) Same as in **B**, but for adaptive binning. Note sparseness of very large avalanches (cp. **B**). (**F**) Adaptive binning increases avalanche rate significantly during LATE while reducing avalanche size (cp. **C**).

DOI: https://doi.org/10.7554/eLife.27119.003

presentation, when the monkey was preparing to reach for the left feeder (*Figure 1D*, single electrode; *Figure 1E*, all electrodes). A much weaker, more delayed increase was observed for right trials. Based on the delayed increase in nLFP rate for left trials, we divided trials into a baseline period before cue-onset (BASE; –0.4–0 s), and an early (EARLY; 0–0.4 s) and late epoch (LATE; 0.4–0.8 s) after cue-onset.

To study neuronal avalanches, two 1 hr recording sessions from 2 consecutive days resulting in 322 trials (day 1: 139, day 2: 183) were combined. In short, for each trial successive nLFPs on the array were concatenated into contiguous clusters at temporal resolution Δt (*Beggs and Plenz, 2003*). A cluster started with the transition from an empty time bin of width Δt to a time bin with at least one nLFP on the array. The concatenation continued until a time bin with no nLFP was encountered (see also *Figure 2A*). The temporal resolution Δt was defined by the inverse of the average nLFP rate on the array and is not a free parameter (*Beggs and Plenz, 2003*). In line with standard avalanche analysis, we kept Δt fixed ('fixed binning') and determined its value from the full recording to be Δt = 4.75 ms for monkey A. We found that the probability density function of cluster size *s*, that is the number of nLFPs in a cluster, followed a power law, for both left and right trials, with slope α close to –1.5 and a cut-off for *s* > 96, the maximal number of MEA electrodes (*Klaus et al.,*

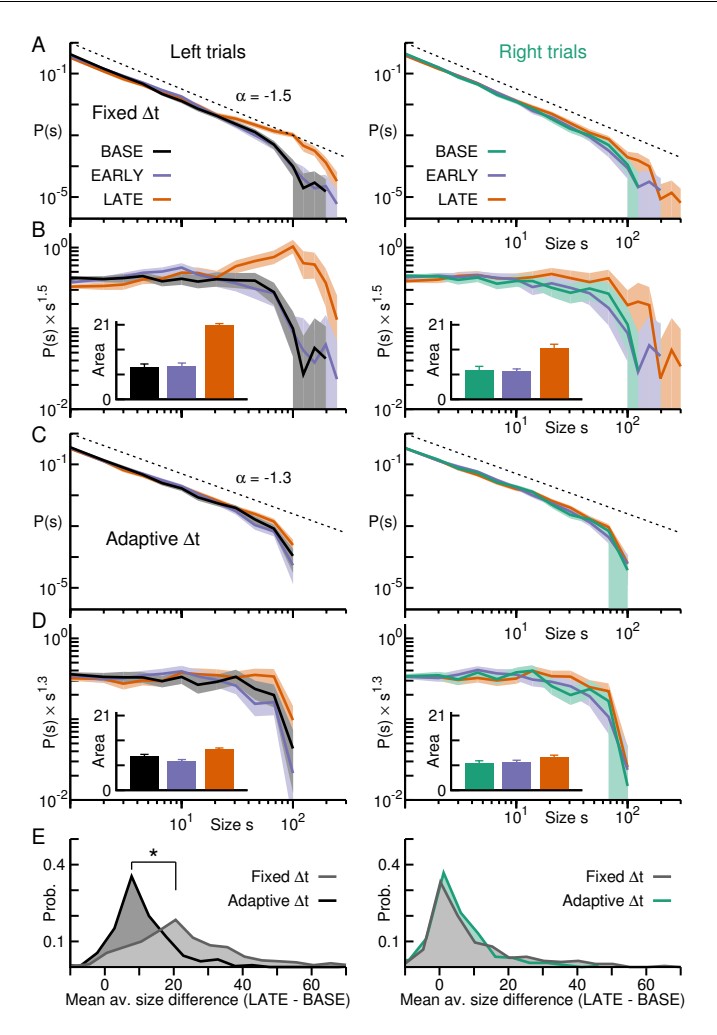

**Figure 3.** Adaptive binning demonstrates maintained power law organization during transient activity epochs in prefrontal cortex. (**A**) Avalanche size probability densities for LATE (*orange*) significantly deviate from the power law observed during BASE (*black/green*) and EARLY (*purple*) for left (*left panel*), but not right trials (*right panel*). *Area*: 95% confidence interval. (**B**) Size distributions normalized by a power law with exponent −1.5 reveal good agreement with theory (horizontal curve with a sharp cut-off), except during LATE for left trials, where a large deviation can be observed. *Inset*: Area under the curve of the normalized distributions for avalanches of size > 20 quantify differences between epochs. *Error bars*: SD from 1000 bootstraps. (**C**) Same as in **A**, but for adaptive binning, which collapses size distributions for BASE, EARLY and LATE. (**D**) Adaptive binning leads to a decrease in the area difference between LATE and the other epochs. (**E**) Distributions of the difference between the mean avalanche size of LATE and BASE for each trial (positive values indicate avalanches in LATE were larger than those in BASE for a given trial) for fixed (*grey*) and adaptive binning (*black/green*). Significant differences between distributions are marked by an asterisk (Kolmogorov-Smirnov test; $p \leq 0.05$). For left trials, LATE is significantly biased toward smaller avalanches for adaptive binning compared to fixed binning.

DOI: https://doi.org/10.7554/eLife.27119.004

*2011*; *Yu et al., 2014*) (*Figure 1F*, solid black and green lines for left and right trials, respectively; exponential vs. power law: $p<10^{-5}$; see Materials and methods). Importantly, these hallmarks of avalanche dynamics were destroyed by trial shuffling, which removes spatial trial-to-trial correlations while maintaining average rate changes (*Figure 1F*, broken lines; exponential vs. power law: $1 - p<10^{-5}$). This control demonstrates that the power law in cluster sizes, indicative of avalanche dynamics, does not simply arise from transient activity changes during behaviorally relevant periods.

By combining clusters from different behavioral epochs into one single distribution, however, transient deviations from scale-invariance could cancel each other, giving the appearance of maintained avalanche dynamics during behavior. Indeed, when separating clusters according to trial epochs BASE, EARLY and LATE, systematic differences in the corresponding avalanche raster were revealed. Specifically, the increase in nLFP rate during LATE for left trials yielded significantly larger avalanches during the period prior to reach (*Figure 2B*, red dots; *Figure 2C*, orange and purple segments indicate significantly high and low values, respectively; $p \leq 0.05$; see Materials and methods). Accordingly, a preponderance of large avalanches was visible in the corresponding size distribution during LATE for left trials (*Figure 3A and B*, left), whereas size distributions remained close to a power law during BASE and EARLY and for right trials (*Figure 3A and B*; $p<10^{-5}$, all cases). In the case of right trials, where the transient changes in activity rate are much less pronounced, size distributions are more similar among all epochs, with a small tendency for larger avalanches during the LATE epoch (*Figure 3A and B*, right; $p<10^{-5}$, all cases).

These deviations from scale-invariance compared to BASE, on the other hand, were abated when the temporal resolution $\Delta t$ to define clusters was obtained from the average nLFP rate for each trial $i$ and epoch separately ($\Delta t_i^{BASE,EARLY,LATE}$; *Figure 2D–F*). We define this approach as 'adaptive binning'. It decreases $\Delta t$ during periods of high nLFP rate, while increasing $\Delta t$ for low activity periods (*Figure 2D*). Indeed, adaptive binning obtained scale-invariant power laws in avalanche sizes that were similar for each epoch (*Figure 3C,D*; $p = 0.06$ $(0.08)$ and $p = 0.11$ $(0.1)$ for EARLY (LATE) compared to BASE, for left and right trials respectively; see Materials and methods), with a change in exponent consistent with previous results (*Beggs and Plenz, 2003*; *Petermann et al., 2009*; *Priesemann et al., 2014*) and a cut-off for $s > 96$ (*Klaus et al., 2011*; *Yu et al., 2014*). Comparing BASE and LATE avalanches for each trial, the bias toward larger ones in the latter epoch seen when employing fixed binning is significantly decreased after adaptive binning for left trials (*Figure 3E*, left), while during right trials, in which scale-free avalanches were obtained even for fixed binning, the bias does not change after adaptive binning. As can be seen in the corresponding raster plots and trial averages (*Figure 2E,F*), avalanche size decreased in LATE concomitant with an increase in avalanche rate (more prominently for left trials). We conclude that the apparent over-representations of large avalanches during behavioral epochs were due to a mismatched temporal resolution $\Delta t$ by not accounting for systematic changes in nLFP rate during different epochs.

We further explored if nLFP rate within an epoch correlates with behavioral outcome. Indeed, reaction time, defined as the time it took the monkey to reach the reward after the visual cue was switched off, did not correlate with nLFP rate (left trials: R = 0.036, p=0.73; right trials: R = −0.039, p=0.71; p-values indicate the likelihood of obtaining a correlation at least as high by chance), avalanche rate (left trials: R = 0.041, p=0.68 for fixed $\Delta t$ and R = −0.022, p=0.79 for adaptive $\Delta t$; right trials: R = −0.033, p=0.74 for fixed $\Delta t$ and R = 0.012, p=0.87 for adaptive $\Delta t$) or avalanche size (left trials: R = 0.085, p=0.52 for fixed $\Delta t$ and R = 0.062, p=0.66 for adaptive $\Delta t$; right trials: R = 0.058, p=0.63 for fixed $\Delta t$ and R = −0.025, p=0.72 for adaptive $\Delta t$), in line with our finding of a scale-free regime regardless of rates for each trial. Given the monkey's high performance (>95% successful trials), rates could not be analyzed for correct/incorrect trial outcome.

## Avalanche maintenance during self-initiated movements

In monkey B, we studied activity changes during self-initiated movements. The monkey, without further cue or prompting, initiated the touching of a pad with her right hand, after which a reward was provided. Similar to monkey A, two 1 hr sessions were recorded in 2 consecutive days. Since results were highly consistent between different days, the data were pooled together, resulting in 663 trials analyzed (day 1: 319; day 2: 344). As the recording site was within the arm representation area within the PM, recorded activity clearly reflected task execution (*Figure 4*). About 200 ms before touching the pad, 18/43 (~42%) putative single-units increased in firing rate, whereas 5/43 units (~12%) decreased their firing rate around the time of touching (*Figure 4A,B*). Fano Factor of unit activity decreased in line with previous reports (*Figure 4C*) (*Churchland et al., 2010*). Similarly, the LFP transiently turned negative ~200 ms before touching for all electrodes (*Figure 4D*), together with a ~4-fold increase in the nLFP rate and a slight decrease thereafter (*Figure 4E*), which readily separated activity into a baseline epoch (BASE; –0.8 to –0.4 s relative to the touch), followed by a pre-touch (PRE; –0.4 to 0 s) and post-touch epoch (POST; 0 to 0.4 s) (*Figure 4F*).

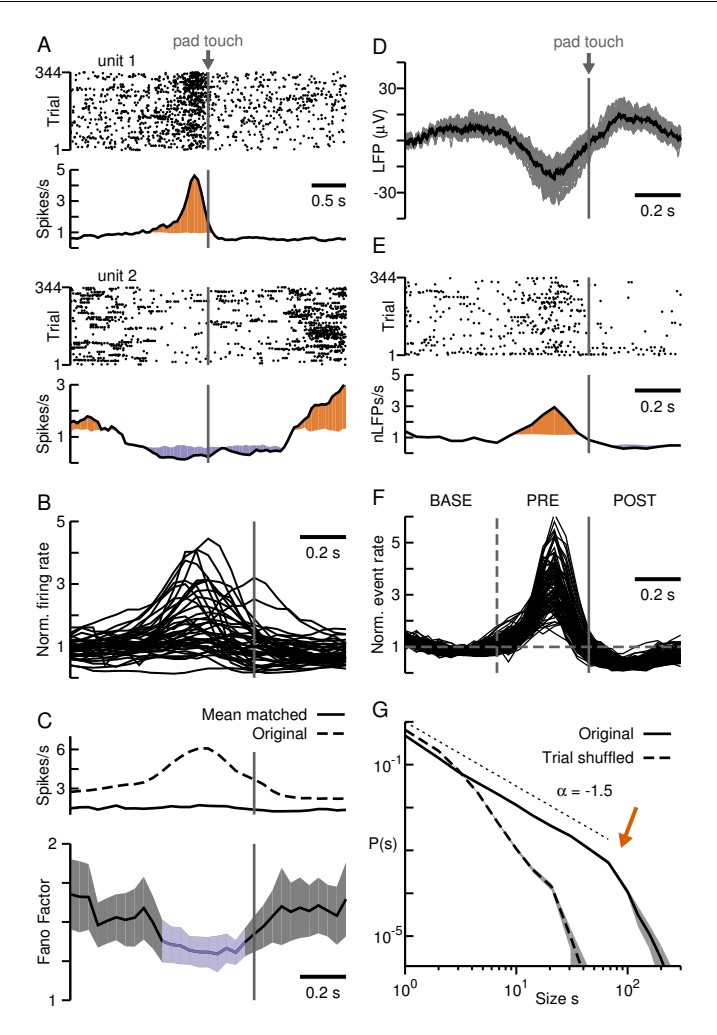

**Figure 4.** Neuronal avalanches in premotor cortex during self-initiated motor task. (**A**) Change in firing rate for two putative single units demonstrate PM region recorded by the array is involved in task performance. Unit one increased whereas unit two decreased firing around the time of self-initiated touch (*grey arrow*). Unit raster (*top*) trial-aligned to touch (*grey line, arrow*) and corresponding average firing rate time course (*bottom*). *Orange/purple* area indicate significant increase/decrease from baseline. (**B**) Most units show transient excitation before touching. Average firing rate change for all putative single units normalized to baseline. (**C**) Average (*top, broken*) and mean-matched (*top, solid*) firing rate across all units and trials. Fano Factor (*bottom*) decreases significantly (*purple*) before touching (70% of data survived mean-matching). *Shaded area*: 95% confidence interval. (**D**) Negative deflection in the LFP, averaged over trials, prior to touch. *Grey*: trial average for individual electrodes. *Black*: average over electrodes. (**E**) Single electrode nLFP raster and corresponding average nLFP rate. *Orange/purple* areas indicate significantly high/low periods from baseline. (**F**) Change in nLFP rate defines baseline (BASE), pre-touch (PRE) and post-touch (POST) epochs (*vertical lines*). (**G**) nLFPs on the array when analyzed for full trial periods show avalanche organization. Power law in size probability densities for nLFP clusters (*solid*; fixed Δt). *Arrow*: Cut-off at number of electrodes on the array. *Broken lines*: Corresponding trial-shuffle control. *Shaded areas*: Corresponding 95% confidence interval based on bootstrapping. *Dotted line*: Visual guide for exponent of −1.5.

DOI: https://doi.org/10.7554/eLife.27119.005

When all data across epochs was analyzed using fixed binning, that is Δt based on the full recording (Δt = 10.3 ms), avalanche sizes followed a power law with –1.5 exponent and cut-off at ~96 electrodes (*Figure 4G*; exponential vs. power law: $p<10^{-5}$), which was destroyed by trial shuffling (exponential vs. power law: $1-p<10^{-5}$), again demonstrating that scale-invariant LFP clusters were

based on the trial-by-trial correlations between cortical sites and did not simply arise from transient changes in nLFP rate.

Evaluation of avalanche raster plots based on epochs revealed systematic differences in the corresponding avalanche raster (*Figure 5A,B*). Specifically, the increase in nLFP rate during PRE yielded significantly more avalanches and of larger size during that period, whereas the converse was true during POST when nLFP rate decreased (*Figure 5A*; colored dots; *Figure 5B*; orange and purple segments respectively; $p \leq 0.05$; see Materials and methods). The corresponding size distributions exhibited deviations from scale-invariance obtained for BASE (*Figure 6A and B*, left) with an excess of large avalanches for PRE and a deficit of large avalanches for POST, despite all distributions remaining close to power laws (exponential vs. power law: $p<10^{-5}$, all three epochs).

As in the case for the cognitive task, these systematic deviations in size distributions were abated when calculating $\Delta t$ from the average nLFP rate for each trial $i$ and epoch ($\Delta t_i^{BASE,PRE,POST}$; *Figure 6A and B*, right). Through adaptive binning we once again obtained scale-invariant power laws in avalanche size that were similar for each epoch ($p = 0.177$, PRE vs. BASE; $p = 0.05$, POST vs. BASE), with the bias towards larger avalanches during PRE being significantly reduced (*Figure 6C*). As can be seen in the corresponding raster plots and trial averages (*Figure 5C,D*), avalanche size decreased in PRE concomitant with an increase in avalanche rate confirming our findings in monkey A. During POST, under representation of large avalanches due to mismatched temporal resolution $\Delta t$ can also be partially compensated for by adaptive binning.

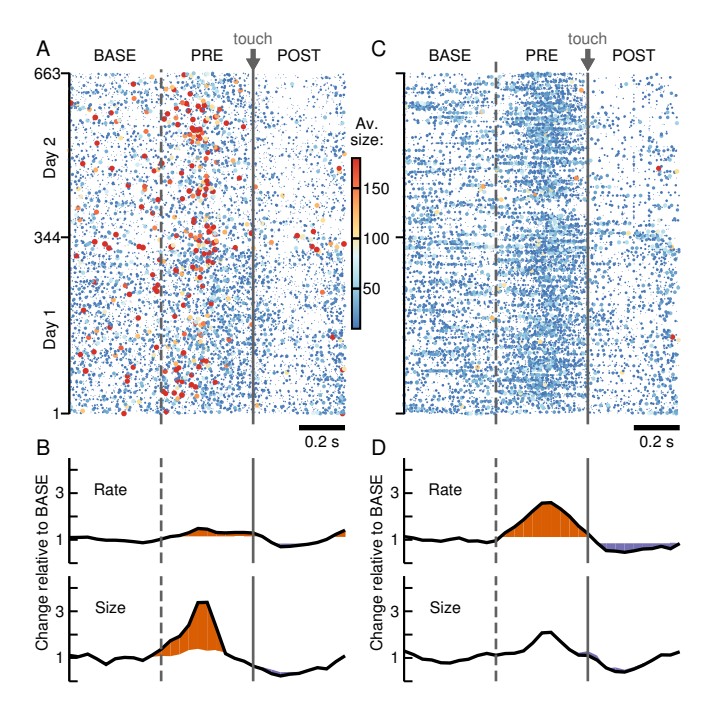

**Figure 5.** Adaptive binning tracks avalanche organization during self-initiated motor task in premotor cortex. (**A**) Avalanche raster with size coded by color and dot size for fixed $\Delta t$. Large avalanches (*red*) emerge during PRE and fewer avalanches are found during POST. (**B**) Significant increases (*orange*) and decreases (*purple*) in average time course for avalanche rate (*top*) and size (*bottom*) with respect to BASE. (**C**) Same as in **A**, but for adaptive binning. Note absence of very large avalanches during PRE. (**D**) Adaptive binning reduces avalanche size while increasing avalanche rate (cp. **B**) during PRE.
DOI: https://doi.org/10.7554/eLife.27119.006

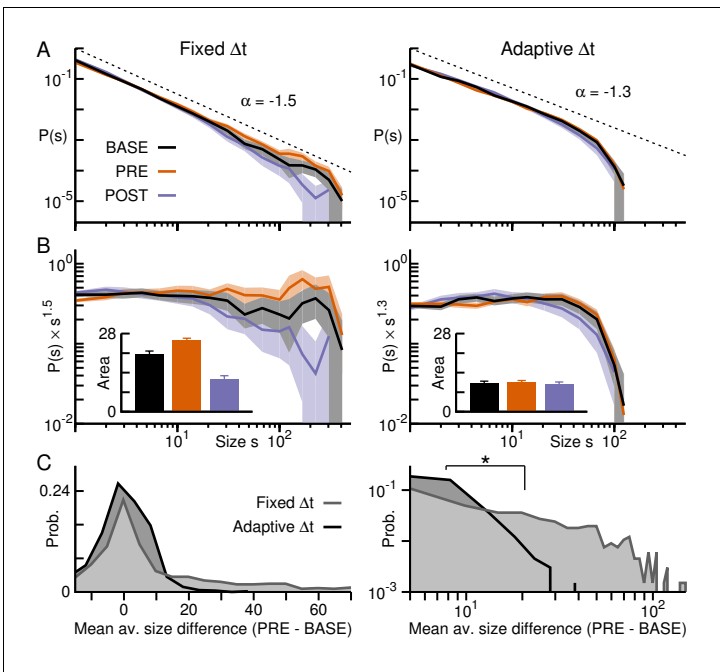

**Figure 6.** Adaptive binning demonstrates maintained power law organization during transient activity epochs in premotor cortex. (**A**) Avalanche size distributions for PRE and POST deviated from the power law observed during BASE when fixed binning was employed (*left*), but collapse together after adaptive binning (*right*). *Area*: 95% confidence interval. (**B**) Size distributions normalized by a power law (exponent −1.5 for fixed and −1.3 for adaptive binning). *Inset:* Area under the curve for the normalized distributions, considering avalanches of size > 20, emphasize the bias toward larger (smaller) avalanches during PRE (POST), in line with higher (lower) activity rates observed during that epoch compared to BASE for fixed binning (*left*). The bias is significantly decreased after adaptive binning (*right*). *Error bars*: SD from 1000 bootstraps. (**C**) Distributions of the difference between the mean avalanche size of PRE and BASE during each trial for fixed (*grey*) and adaptive binning (*black*). Large differences occur much more often for fixed binning (*left:* linear-linear plot; *right:* log-log plot). Significant differences between distributions are marked by an asterisk (Kolmogorov-Smirnov test; $p \leq 0.05$).
DOI: https://doi.org/10.7554/eLife.27119.007

## Impact of 'adaptive binning' in capturing rate change vs. change in dynamics

We next illustrate the expected deviation from scale-invariance when rates change and its remediation using 'adaptive binning' in a simple schematic. In the hypothetical example shown in *Figure 7A*, we consider a baseline period with nLFP rate of 3 Hz per electrode, followed by a period with the rate increased to 9 Hz (*Figure 7A*, left). Fixed binning, which sets Δt according to the average rate of 6 Hz, underestimates the temporal resolution for baseline activity, whereas it overestimates Δt for the high activity period. The resulting power law size distributions will be steeper for the baseline period and shallower for the high activity period (*Beggs and Plenz, 2003*). The combined distribution will show an upward bend at the cross-over of the individual distributions (*Figure 7A*; middle, solid line) which seems to suggest a deviation from avalanche dynamics with an overabundance of large clusters similar to what would be expected for supercritical dynamics. Adaptive binning, that is calculating Δt for each period separately approximates the exponents for both power laws, resulting in a final distribution that matches a power law with an intermediate slope (*Figure 7A*, right) (*Beggs and Plenz, 2003*; *Petermann et al., 2009*; *Priesemann et al., 2014*). We note that simply using shorter time bins for all data would retain different slopes for different regimes and thus is not an alternative to adaptive binning.

Although we illustrated this principle using two sequential periods with different rates, the same holds for repeat trials with different rates. In *Figure 7B*, we separated trials of monkey B into low- and high-rate responses based on PRE epochs (*Figure 7B*, left). Size distributions obtained from

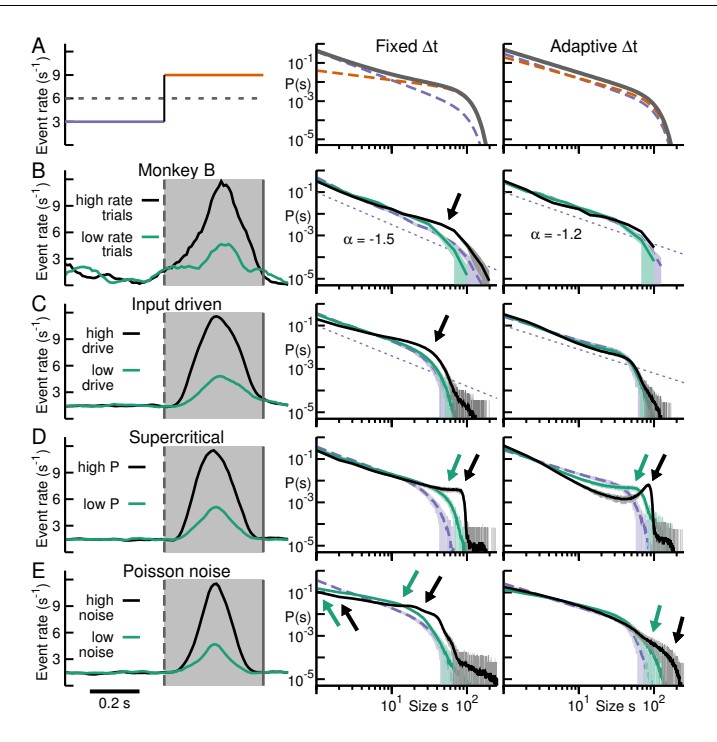

**Figure 7.** Adaptive binning recovers the power law in the face of consistent rate changes when dynamics remain critical. (**A**) Schematic impact of avalanche analysis from non-stationary event rates. *Left*: A low event rate period (*purple*) followed by a high event rate period (*orange*) results in an intermediate $\Delta t$ (fixed binning) based on the mean event rate (*broken line*). *Middle*: The superposition of two power laws with different slopes from their corresponding rate regime (*broken lines*) results in an avalanche size distribution (*grey*) that deviates from a power law, with a characteristic up-ward bend at the cross-over point. *Right*: Adaptive binning steepens/reduces the slope for the high/low rate period respectively resulting in a distribution collapse at an intermediate slope. (**B**) *Left*: Average nLFP rate for monkey B with trials separated into high (*black*) and low (*green*) nLFP rate during PRE (*shaded area*). *Middle*: Corresponding distributions obtained with fixed binning increasingly deviate with rate (*arrow*) from baseline (*broken, purple*). *Right*: Adaptive binning collapses all distributions. (**C**) Simulations using a transient external Poisson drive match experimental findings (cp. **B**). (**D**) In simulations with transiently switching from critical to supercritical dynamics (*left*), adaptive binning fails to compensate for overabundance of large avalanche sizes from supercritical dynamics (*middle/right*). (**E**) Simulations using transient Poisson noise. Distributions obtained with fixed binning do not follow power laws, even in the low-noise regime (*middle*). Distributions obtained with adaptive binning do not have a clear cut-off (*right*).
DOI: https://doi.org/10.7554/eLife.27119.008

high-rate trials at fixed binning exhibit an overabundance of large avalanches (*Figure 7B*, middle, arrow). With adaptive binning, deviations from the baseline distribution decreased and all distributions approached power laws with a −1.2 exponent (*Figure 7B*, right). This demonstrates that trials in monkey B exhibit similar dynamics as those observed for monkey A or, more generally, that deviations from scale-invariance observed in the two experimental regimes can be explained by nLFP rate changes alone.

## Neuronal simulations support 'adaptive' binning to recover scale-invariance for critical dynamics

Using simulations, we explored the conditions under which adaptive binning recovers scale-invariant size distributions. Simulations were carried out on a $10 \times 10$ cellular automaton network, the approximate size of our MEAs, with cascades unfolding according to a critical branching process (*Zapperi et al., 1995*) (see Materials and methods). Quiescent times between cascades were randomly drawn from the experimentally obtained quiescent time distribution of monkey B. Simulated baseline activity, by randomly initiating avalanches in the network, was superimposed after 0.4 s by a

second process, which produced a transient increase in rate mimicking movement initiation. We tested three different processes for two levels of rate increase and obtained corresponding size distributions for fixed and adaptive binning. Increasing rate by adding Poisson inputs that could trigger avalanches (*Figure 7C*, 'Input driven') closely matched our experimental results. For fixed binning, an overabundance of large avalanches close to the cut-off was found in the size distribution (*Figure 7C*, middle, arrow), which originated from the concatenation of spontaneous with input-triggered avalanches. We note that this overabundance manifests while the model interactions remained tuned to the critical point in the limit of zero input. Accordingly, the power law in the size distribution was recovered by means of adaptive binning (*Figure 7C*, right). Next, a high rate was achieved by increasing the likelihood of nodes exciting each other, that is changing to a supercritical branching process with increased synchronization (*Figure 7D*, 'Supercritical'). The corresponding size distribution exhibits an overabundance of large avalanches that (*Figure 7D*, middle) reflects explosive growth as activity propagates in the network. Adaptive binning, which is tied to the overall increase in event rate, was insufficient to compensate for the increase in synchrony. It failed to collapse the cut-off of the distributions, instead further separating non-global cascades from synchronized, global activity (*Figure 7D*, right, arrows). Finally, event rate was increased by adding uncorrelated activity, that is Poisson inputs that do not trigger avalanches (*Figure 7E*, 'Poisson noise'). The resulting size distribution exhibits a preponderance of intermediate size avalanches that reflects the mean rate of uncorrelated events introduced (*Figure 7E*, middle). Here, even low-noise levels destroy the power-law regime. While distributions tend to get closer to a power law with adaptive binning, their cut-off at system size is lost in the process (*Figure 7E*, right).

## 'Adaptive' thresholding as an alternative approach to 'adaptive' binning

Adaptive binning reduces $\Delta t$ for periods of high rates, thereby reducing the number of nLFPs per time bin. Alternatively, one can increase the threshold for nLFP detection during these periods, likewise reducing the number of nLFPs encountered per time bin (see also [*Petermann et al., 2009*]). The reverse argument holds when periods of low rates are encountered. We tracked the mean and standard deviation of the LFP in successive windows of 100 ms width and used a moving threshold $thr$ = mean – 2SD to identify nLFPs within each window (*Figure 8A,C*). Thus, activity modulations induced by the task were mainly reflected in nLFP amplitude, but not rate. This approach, which trades temporal resolution for local sensitivity, collapsed distributions from different periods (grey and orange areas in *Figure 8B,C*) in monkey A into a similar power law, indicative of sustained avalanche dynamics during task performance (*Figure 8D,E*).

## Phase synchrony analysis supports recovered scale-invariance through 'adaptive' binning

Experiments and theory have consistently shown that size distributions of clustered activity are sensitive to the degree of synchrony in a system (*Shew et al., 2009*; *Larremore et al., 2011*; *Shew et al., 2011*; *Yang et al., 2012*; *Bellay et al., 2015*; *Shew et al., 2015*). Our demonstration of scale-invariant size distributions suggests avalanche dynamics are maintained throughout task performance, which should mirror a sustained average synchrony among cortical sites. Previous studies have shown overall phase-synchronization between sites sensitively captures fast dynamical changes in a network (*Kitzbichler et al., 2009*; *Yang et al., 2012*), in particular at higher frequencies (*Meisel et al., 2016*). In line with our results from adaptive binning and thresholding, levels of LFP synchrony $R$ (see Materials and methods) were not different across epochs in monkey A (Left: 0.56 ± 0.015, 0.56 ± 0.01 and 0.56 ± 0.01; Right: 0.56 ± 0.01, 0.55 ± 0.01 and 0.56 ± 0.004; mean ± standard deviation for BASE, EARLY and LATE respectively; $p>0.05$; two-tailed paired student's t-test) as well as in monkey B (BASE, 0.450 ± 0.003; mean ± standard deviation; vs. PRE, 0.452 ± 0.003; vs. POST, 0.455 ± 0.003; $p>0.05$).

## Discussion

Our findings suggest that cortical avalanches are maintained in nonhuman primates during information processing, thus adding avalanche dynamics to the many links found between ongoing and evoked activities in the brain (*Arieli et al., 1996*; *Tsodyks et al., 1999*; *Womelsdorf et al., 2006*).

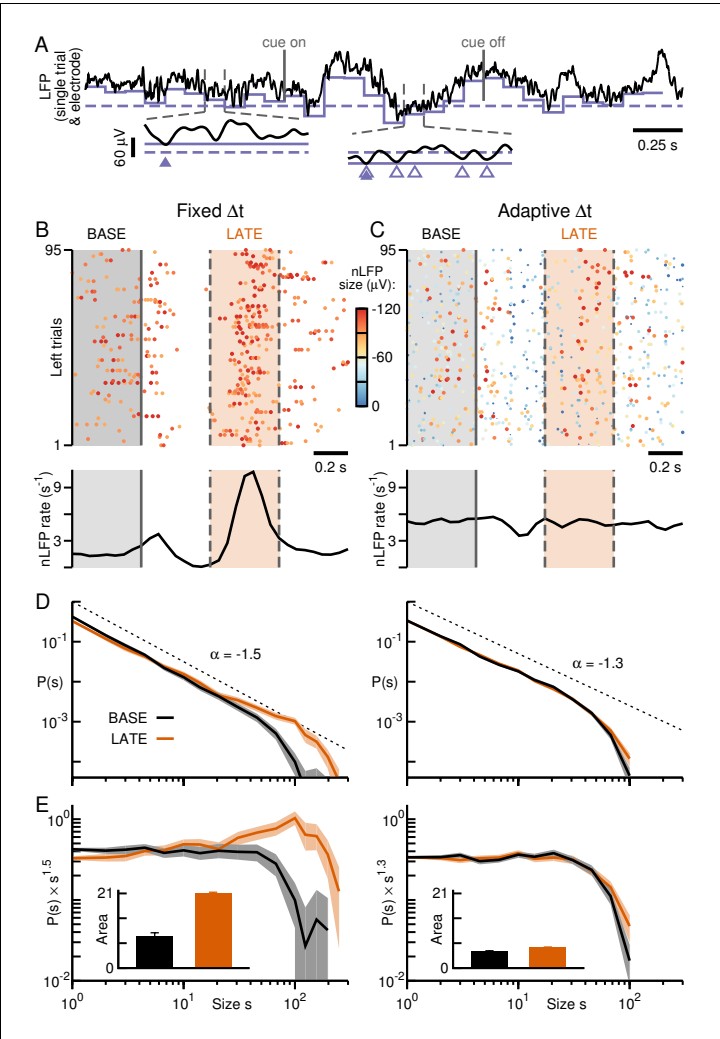

**Figure 8.** Adaptive thresholding as an alternative means to collapse size distributions during rate changes. (A) Example LFP on single electrode in monkey A. *Solid grey lines*: Cue presentation. nLFPs detected at fixed (*broken, purple*) or adaptive threshold (*solid, purple*) vary in size and rate accordingly. *Triangles:* nLFPs obtained with fixed (*empty*) or adaptive thresholding (*filled*). (B) nLFP raster (*top*) and corresponding average nLFP rate (*bottom*) from A at fixed threshold. *Color code*: nLFP amplitude in μV. (C) Adaptive thresholding produces a relatively constant event rate (cp. B). (D) Adaptive thresholding (*right*) successfully collapses avalanche size distributions obtained for fixed thresholding (*left*; cp. *Figure 3A and C*, left). (E) Distributions normalized by a power law of −1.5 (fixed thresholding, *left*) or −1.3 (adaptive thresholding, *right*) together with area differences are in line with the results from adaptive binning (cp. *Figure 3B and D*, left). *Error bars*: SD from 1000 bootstraps.
DOI: https://doi.org/10.7554/eLife.27119.009

We suggest that optimal information processing capabilities of avalanche dynamics (*Beggs and Plenz, 2003*; *Bertschinger and Natschläger, 2004*; *Haldeman and Beggs, 2005*; *Kinouchi and Copelli, 2006*; *Rämö et al., 2007*; *Nykter et al., 2008*; *Shew et al., 2009*; *Tomen et al., 2014*; *Gautam et al., 2015*) might guide sensory, motor and cognitive processing in the brain.

During ongoing activity, avalanche dynamics have been demonstrated in various brain areas including the primary visual (*Hahn et al., 2010*), primary motor (*Petermann et al., 2009*), somato-sensory (*Gireesh and Plenz, 2008*), premotor and prefrontal (*Yu et al., 2014*) cortices, reflecting a common dynamic feature regardless of specific cortical areas. Therefore, we aimed to extract a simi-lar rule that govern avalanches during evoked activity. To achieve this goal while minimizing the number of nonhuman primates used, we chose to record activity from different cortical sites and dif-ferent behavioral tasks in two monkeys, while the common condition is that the recorded region was

strongly activated by the corresponding task. Monkey A performed a visual-motor mapping task, in which the correct actions have to be applied to specific cues (*Wise and Murray, 2000*). This task is known to involve the dorsal-lateral prefrontal cortex (*Asaad et al., 1998*; *Puig and Miller, 2012*), in line with our finding of differential spiking as well as LFP responses for the left and right cues. Conversely, the target area in our self-initiated motor task for monkey B was the premotor cortex, which is involved in movement planning (*Weinrich et al., 1984*; *Brasted and Wise, 2004*) and self-initiated movement (*Hoffstaedter et al., 2013*). Consistent with previous studies, we found that, at both the single neuron and population level, the premotor cortex exhibited strong activation a few hundred milliseconds before the self-initiated touch. Our highly consistent results obtained for these different cortical areas and behavioral tasks suggest that avalanche dynamics might be a general principle governing cortical dynamics underlying various behaviors.

Our trial-shuffling control demonstrated that the power law in avalanche sizes reflects correlations that are maintained during behavior on a trial-by-trial basis and that this statistical feature did not simply arise from non-stationary rates. Recently, power laws, for example Zipf's law, have been shown to arise when the range of event probabilities becomes enlarged, for example from underlying common correlations due to common inputs typically captured in latent variables (*Schwab et al., 2014*; *Aitchison et al., 2016*). Because trial-shuffling maintains such latent variables as captured in the average evoked response, yet, abolishes scale-invariance, our findings suggest that the scale-invariance encountered in our behavioral data reflects intrinsic correlation structure of trial-to-trial variability and not common correlations.

Originally introduced for resting or spontaneous activity (*Beggs and Plenz, 2003*), a fixed temporal resolution Δt assumes stationary rates, which is typically violated during sensory epochs or periods of movement when event rates change systematically. For avalanche analysis, the temporal resolution Δt is based on the inter-event interval distribution of the population, for example successive nLFP events on the array. This temporal scale Δt is not a free parameter. It further depends on the spatial scale, that is inter-electrode distance Δd, as well as local minimal event threshold *thr*. Importantly, all three parameters Δt, Δd and *thr* have been experimentally linked and shown to affect the slope of the power law in a predictable manner (*Beggs and Plenz, 2003*; *Petermann et al., 2009*). In our initial approach, that is adaptive binning, we kept the threshold *thr* and inter electrode distance Δd constant while changing Δt in accordance with the change in event rate, which in our experiments increased up to a factor of 10. This recovered power law statistics, which was confirmed by our simulation in which rate changes were created by increasing avalanche rate. These findings are in line with reported changes of the power law slope with changes in Δt (*Beggs and Plenz, 2003*; *Petermann et al., 2009*; *Priesemann et al., 2014*).

Our results highlight the importance of the principle of 'separation of time scales' (*Bak et al., 1988*; *Vespignani and Zapperi, 1998*; *Plenz, 2012*; *Priesemann et al., 2014*) for measuring avalanche dynamics. That is, the time scale at which cascading events unfold within individual avalanches, characterized by the avalanche duration, should be significantly shorter than the time scale at which different avalanches emerge, characterized by the inter-avalanche interval. Estimating the temporal resolution Δt from the inverse of the event rate provides a reasonable separation of time scales, because it balances two potential errors – false concatenation of separate avalanches and false separation of a single avalanche (*Beggs and Plenz, 2003*; *Plenz, 2012*).

In an alternative approach, we kept spatial resolution Δd and temporal resolution Δt fixed while changing threshold *thr*. This approach does not affect the power law organization, as demonstrated in nonhuman primates (*Petermann et al., 2009*) and human fMRI (*Tagliazucchi et al., 2012*). Particularly when Δt is difficult to change, changing *thr* might provide an alternative means to reestablish a separation of time scales, since large local events are less common than small local events. Our recovery of a power law for large local events during high activity periods demonstrates the analogous roles time and local event size play in avalanche dynamics. Our separate demonstration that synchrony does not change between BASE, EARLY/PRE and LATE/POST conditions further supports our finding that avalanches are maintained despite changes in event rates. Conversely, if thresholds are calculated individually for different temporal segments corresponding to different conditions of interest and the resulting distributions deviate from power laws along with concurrent changes in synchrony measures, then one has strong indication that the underlying dynamics deviate from criticality (*Meisel et al., 2013*).

It is important to emphasize that adaptive binning and thresholding are only tools to reveal the signatures of scale-free dynamics after they become blurred by increased drive and absence of separation of time-scales. They are not meant to be a perfect theoretical solution to the problem and have obvious limitations. For instance, there is no clear answer as to how large the window for calculating adaptive binning or thresholding should be. If it is too small, the inherent fluctuations of a critical system will be regarded as rate changes and therefore no power law dynamics can be observed. If it is too large, the rate changes can be averaged out inside them, and therefore the problem of the non-stationarity would remain. In the case of our experiments, we had a clear indication of when we should expect a rate change, which gave us a good parameter to choose the window size (~400 ms). Another problem arrives for extreme rate regimes. For very high drive, in which Δt would be much shorter than the time it takes for spikes to propagate from a neuron to its neighbors, it is unlikely that a power law can be recovered since activity becomes uncorrelated at that temporal resolution. On the other extreme, if there is almost no activity to be observed, a very large bin may require unrealistic long recordings to produce power laws. These two extreme cases may explain some of the residual error we found for left trials during PRE for monkey A (slightly larger avalanches even after adaptive binning) and during POST for monkey B (slightly smaller avalanches even after adaptive binning).

Another important point is that the advantages of criticality within a certain area allow it to optimally process information, regardless of the nature of the input received by that area. In fact, one of the advantages of criticality is maximal dynamic range (*Kinouchi and Copelli, 2006*; *Shew et al., 2009*; *Gautam et al., 2015*), which means that the downstream network which employs criticality will be able to respond properly for a larger range of different inputs. Nevertheless, it is possible downstream neurons employ a sort of adaptive binning mechanism. It has been suggested in simulations that the activity of an upstream network can influence how individual downstream neurons perform spatial and temporal integration at the synaptic level (*Bernander et al., 1991*; *Rapp et al., 1992*). The more input a cell receives, the more synchronized these incoming spikes have to be in order to produce a spike on the post-synaptic cell, effectively increasing the temporal resolution by which this downstream cell process its input (*Bernander et al., 1991*), similarly to how our proposed adaptive binning method works.

Neuronal avalanche dynamics are predominantly located in cortical layers 2/3 (*Stewart and Plenz, 2006*; *Petermann et al., 2009*) and exhibit strong non-linear components involving the interaction between cortical sites, that is the interactions of neurons or neuronal groups (*Thiagarajan et al., 2010*; *Yu et al., 2011*; *Plenz, 2012*). Accordingly, maintained avalanche dynamics during self-initiated and cue-related responses suggest non-linear interactions in superficial layers during cortical processing, which numerous studies have shown to be indeed the case. For example, evoked responses in superficial layers of visual cortex in the nonhuman primate have been found to be non-linear, potentially involving the local recurrent network (*Williams and Shapley, 2007*; *Xing et al., 2012*). Similarly, functional connectivity based on the interaction from neuronal firing in layer 2/3 of nonhuman primates has been found to contribute to motor coding (*Hatsopoulos et al., 1998*) and to be as robust or even superior over tuning curves in predicting motor outcome (*Stevenson et al., 2012*). With respect to their spatiotemporal spread, avalanches reveal a fractal dimension when analyzed individually (*Plenz, 2012*), with nearest-neighbor relationships in the average (*Gireesh and Plenz, 2008*; *Yu et al., 2014*), in line with compact spatiotemporal spreading of average evoked responses reported, for example, for premotor cortex (*Rubino et al., 2006*).

Our current findings may retroactively explain previous reports that found deviations from avalanche dynamics during stimulus presentation. *Arviv et al. (2015)* found power law avalanche size distribution when considering the full 1 s window for a visual detection task in human MEG recordings. This is consistent with our results when considering the entire period of task performance (~2 s). They also observed a trend toward supercritical dynamics during transient activity periods, just as we found predominantly large avalanches during high rate periods. Our analysis, though, shows adaptive binning as one potential approach to distinguish true supercritical dynamics from avalanche dynamics. *Fagerholm et al. (2015)* reported subcritical dynamics during focused attention in human EEG, in line with the observation of reduced correlations during such episodes (*Cohen and Maunsell, 2009*; *Mitchell et al., 2009*; *Harris and Thiele, 2011*). However, changes in activity levels were not reported in that study, and size distributions were evaluated outside their cut-off (*Yu et al., 2014*). Finally, early stimulus responses in the turtle's visual system (*Shew et al., 2015*) have been

reported to deviate from avalanches, a finding which might reflect the statistics of the stimulus used rather than intrinsic cortical dynamics.

Numerical simulations of systems under external drive or with non-zero spontaneous neuronal activity can never be truly critical, since the quiescent phase (and with it the transition connecting it to an active phase) disappears under these conditions (*Taylor et al., 2013*; *Hartley et al., 2014*; *Williams-García et al., 2014*). This effect is increasingly pronounced in simulations with small network size and large activity drive. Given the relatively small change in firing rate observed during behavior and the large size of the cortical network, our findings are in line with simulations that show weakly driven critical system to exhibit power law behavior when the temporal resolution is matched to the activity rate (*Hartley et al., 2014*).

In conclusion, our findings provide strong evidence that the scale-free organization of neuronal avalanches is maintained during evoked responses in premotor and prefrontal cortex, despite systematic fluctuations in firing rates. We therefore suggest that neuronal information processing during tasks might capitalize on various functional benefits shown for critical dynamics. This calls for further investigation into how critical dynamics may explain the execution of specific brain functions. Of equal importance, our results also shed new, methodological light on how to identify true deviations from avalanche dynamics under pathological conditions.

## Materials and methods

### Behavioral training and electrophysiological setup

All procedures followed the Institute of Laboratory Animal Research (part of the National Research Council of the National Academy of Sciences) guidelines and were approved by the NIMH Animal Care and Use Committee (protocol #LSN-11). Two adult rhesus monkeys (*Macaca mulatta*) were surgically implanted with a titanium head post each under sterile conditions while under isoflurane anesthesia. After recovery, the monkeys were trained to sit head-fixed in a primate chair for behavioral performance. In the cue-initiated task, monkey A (male, 9 years old, 8 kg) had to press a bar in front of the chair upon presentation of the 'trial-initiation' cue on a computer screen (a grey square in the center of the screen). After ~2 s, the initiation cue was followed by an 'instruction' cue, either 'green cross' or 'red circle', for the duration of 1 s. Upon cue disappearance, monkey A had to release the bar and reach with his right arm to one of two specialized feeders (*Mitz et al., 2001*). The 'green cross' instructed the monkey to reach to the left feeder for reward ('left trials'; contralateral to the reaching hand); a 'red circle' instructed reaching for the right feeder ('right trials'; ipsilateral to the reaching hand). Approaching the incorrect feeder rapidly triggered a proximity sensor to sequester the food rewards in both feeders, which prevented the monkey from obtaining a reward on that trial. The inter trial interval was 3–5 s. In the self-initiated motor task, monkey B (female, 8 years old, 7 kg) had to move her right arm to touch a pad placed ~30 cm in front of the monkey chair after which a food reward was given. Pad touching was self-initiated: no cue was presented. After the monkeys learned their respective tasks, a multi-electrode array (MEA; 96 channels - 10 × 10 without corners, inter-electrode distance: 400 μm; BlackRock Microsystems) was chronically implanted in the left prefrontal area (area 46, monkey A; electrode length: 0.55 mm) or the arm representative region of the left premotor cortex (monkey B; electrode length: 1 mm – PM is thicker than PFC, therefore longer shanks were employed). After recovering from the implantation surgery (~1 week), behavioral training resumed. The LFP (1–100 Hz band pass filtered; 2 kHz sampling frequency) and extracellular unit activity (0.3–3 kHz band pass filtered; 30 kHz sampling frequency) were simultaneously obtained from the implanted MEA. Electrophysiological signals as well as the timing of behaviorally relevant events, for example touching the pad, presentation of visual cues, etc., were stored for off-line analysis.

### Unit analysis

Extracellular unit activity was extracted offline by spike sorting (Offline Sorter, Plexon Inc.). Putative single-unit activity was identified whenever a clear clustering and separation of waveforms could be identified in at least one feature space. In total, 29 and 43 putative single-units were identified in monkeys A and B, respectively. For calculating the peristimulus time histogram (PSTH), firing rates were calculated for a 250 ms window moving in steps of 50 ms. Statistical significance of firing rate

changes was based on surrogate spike trains from individual units obtained by randomly permuting inter-event intervals. Specifically, a firing rate change was considered significantly above (below) expectation when it was among the top (bottom) 2.5% of the firing rate distribution obtained from the shuffled data. For this analysis, left and right trials were treated separately for monkey A.

## Fano Factor

We employed the method described by *Churchland et al. (2010)* to calculate the Fano Factor, defined as the variance (over trials) of the spike count divided by the mean, in order to evaluate how the variability of the neuronal population studied evolved with time. We computed the variance and mean spike count for each time window (100 ms sliding window moving in steps of 40 ms) for each neuron separately. After that, a linear regression was performed on the scatterplot of the variance versus the mean spike count in which each point represents a single neuron. The slope of this regression measures the Fano Factor for the relevant time window, and the estimated error in the slope calculation provides a 95% confidence interval. To account for the effect from the increase in firing rate on the Fano Factor, a mean-match procedure was employed as described previously (*Churchland et al., 2010*). In short, this procedure removes units from the analysis until the mean firing rate of the remaining units does not change significantly between the periods studied (see *Figures 1B* and *4C*; the amount of data surviving the respective mean-matching is given in the legends). Statistical significance of changes in the Fano Factor due to the task was assessed by a p-value computed from the probability of observing a given slope based on the confidence interval calculated from the baseline level.

## LFP analysis

Negative deflections in the LFP (nLFPs) were detected by applying a threshold at –2 (monkey A) or –2.5 (monkey B) SD of the continuous LFP fluctuations estimated for each electrode separately (fixed and adaptive binning; for adaptive thresholding see below). The time stamps of nLFPs, determined by the data points with the largest negative amplitude, were then employed for avalanche analysis. The same procedure employed to assess the significant of firing rates was also employed for nLFP rates.

## Avalanche analysis using fixed and adaptive temporal binning

Avalanches were identified by concatenating nLFPs occurring in successive time bins of width $\Delta t$ on the array into temporally contiguous spatiotemporal clusters as described originally (*Beggs and Plenz, 2003*). The avalanche size was defined as the number of nLFPs within the avalanche. The temporal resolution $\Delta t$ for the avalanche analysis was defined by the inverse of the average nLFP rate on the array over a given period and thus is not a free parameter. Two approaches for binning the data were employed. The first one, fixed binning, was obtained by calculating the average nLFP rate across all epochs (see *Figures 1* and *4* for different epochs). The second one, adaptive binning, was obtained by calculating the average nLFP rate over different epochs separately. Therefore, this second method resulted in $\Delta t$ that varied between epochs according to changes in the activity rate. The adaptive binning method calculates the temporal resolution on a trial-by-trial basis, taking into account highly variable responses across trials.

## Significance of changes in average avalanche rate and size

In order to test for significant changes in average avalanche rate and size during task performance, we compared the values from each point in time to those obtained from 100 shuffled sets, providing a p-value. Significance was defined at $p \leq 0.05$. For size analysis, shuffled sets were constructed by randomly permuting avalanche sizes, while keeping avalanche occurrence time (thus also keeping the avalanche rate unchanged). For rate analysis, shuffled sets were constructed by randomly assigning a new occurrence time (within the same trial, drawn from a uniform distribution) for each avalanche. For monkey A, left and right trials were treated separately.

## Avalanche size distribution analysis

In order to fit power laws and exponentials to the probability densities of avalanche size, and obtain the corresponding exponents, we employed the Maximum Likelihood Estimation method, as

previously described (*Clauset et al., 2009*; *Klaus et al., 2011*). The distribution used for the power law fit had a sharp cut-off: $p(s) = Cs^{-\alpha}\exp[-(s/s_0)^\gamma]$, where the normalization constant is given by $C = 1/\left(\sum_{s=s_{min}}^{s_{max}} s^{-\alpha}\exp[-(s/s_0)^\gamma]\right)$. The three parameters to be determined were the exponent of the power law $\alpha$, the cut-off $s_0$ and the strength of the decay beyond the cut-off $\gamma$. In order to evaluate the uncertainty in the size distributions, we employed a bootstrap procedure (*Efron, 1979*) to estimate the 95% confidence interval for the probability density at each avalanche size from 1000 resampled data sets, which were obtained by randomly sampling the same number of avalanches found in each case from the original distribution. In the case of model distributions (see below), 100 different instances of the network were simulated, from which the confidence intervals were obtained.

The significance of the different fits to the data was obtained by employing the likelihood test: $D = -2[\ln(L_{pl}) - \ln(L_{alt})]$, where $D$ is the test statistics and $L_{pl}$ is the likelihood for the power-law fit. A p-value is obtained using the calculated statistics from the chi-squared distribution. When comparing power laws to exponentials $L_{alt}$ is the likelihood for the exponential fit, and the p-value computes the chances that the fit with lower likelihood value is actually better. When comparing different distributions $L_{alt}$ is the likelihood for the alternative power-law fit (e.g. when comparing PRE distribution to BASE in *Figure 6A* right, the alternative fit is the one obtained for the BASE distribution). In this case, the p-value computes the chances that the compared distribution cannot be explained by the alternative fit or, in other words, how similar they are. Note that a p-value above 0.05 in this case does not imply that the distributions are significantly different (whereas p-values of 0.05 or lower indicate that the distributions are significantly similar).

## Trial shuffling

In order to assess the influence of rate modulation introduced by the task on avalanche size distributions, we compared the original data to data obtained by randomly shuffling the trial order for each electrode independently, that is, the shift predictor (*Gerstein and Perkel, 1969*; *1972*). This procedure created datasets that preserved the average change in nLFP rate for all electrodes and trials, but destroyed spatio-temporal correlations within individual trials.

## Cellular automaton models

To study the interplay between avalanches and activity rate, we employed a network of $10 \times 10$ cellular automata (*Kinouchi and Copelli, 2006*; *Ribeiro et al., 2014*). Each site, which represents the activity of a single channel from the experimental data, cycles through its 11 states: $x_i(t) = 0$ if the $i^{th}$ site is quiescent at time $t$, $x_i(t) = 1$ if it is active, and $x_i(t) = 2, \ldots, 10$ if it is refractory. A quiescent site at time $t$ can become active at $t + 1$ if any of its pre-synaptic neighbors is active at $t$ and transmits successfully, each connection independently with probability $P$. Once a site is active, its state is incremented according to the following equation until it is back to quiescence: $x_i(t + 1) = [x_i(t) + 1] \mod 11$ (deterministic refractory period). Each site sends $k = 16$ connections to randomly chosen post-synaptic sites, thus forming a random network. Each connection has a probability $P$ of transmitting a spike, which was tuned according to $P = 1/k$ in order to achieve critical dynamics in the network. For avalanche initiation, we randomly chose a site to become active in an otherwise quiescent network. After the propagation of activity ended, a time interval drawn from the inter-avalanche interval distribution obtained from experimental data was imposed before another avalanche was initiated.

In this model, the event, that is nLFP, rate was adjusted according to the baseline level in the experimental data by employing a time step of 2 ms (implying a refractory period of 18 ms). Four different approaches to introduce a transient increase in the activity rate, as observed in the recordings from both monkeys, were studied. In the *Input driven* model, we added independent Poisson inputs that triggered avalanches for each site, with a rate that increases and then decreases linearly from trial time –0.35 s to –0.05 s, peaking at two different possible levels: in the low drive regime, $h_{max} = 1\ s^{-1}$ and in the high drive regime, $h_{max} = 6.5\ s^{-1}$. In the *supercritical* model, we increased the probability of transmission $P$ by multiplying it by a modifier with the same temporal profile described above for the drive, with the modifier ranging from 1 (no change) to a maximum of either 1.5 (50% increase in the probability of spike transmission) or 2.5 (150% increase). Note that this led to both

increased activity rate and supercritical dynamics. In the *Poisson noise* model, we added independent Poisson inputs that could not trigger avalanches to each site, with a rate similar to the one employed for the Input driven model, but peaking at either 17.5 s$^{-1}$ (low-noise regime) or 35 s$^{-1}$ (high-noise regime).

For avalanche analysis, the methods used for experimental data were also employed for the model. In order to reduce artificial separations of single avalanches during adaptive binning, that is when Δt becomes smaller than the time step of the model, we introduced a small jitter in event times ($\leq$ half of the 2 ms time-step). This procedure effectively transformed the discrete temporal dynamics of the model into a continuous one. Jittering did not change our results for experimental data or simulations (data not shown).

## Adaptive thresholding

For each electrode, the mean LFP amplitude and SD were calculated for consecutive windows of duration 100 ms. For each window, the adaptive threshold *thr* was set to mean − 2SD.

## Synchronization measures

We derived estimates of mean phase synchronization for band-pass filtered data. After filtering the data (50–100 Hz frequency band; phase neutral filter by applying a second-order Butterworth filter in both directions), we first obtained a phase trace $\theta_i(t)$ from each LFP channel $F_i(t)$ by applying its Hilbert transform H[$F_i(t)$]: $\theta_i(t) = \tan^{-1}(H[F_i(t)]/F_i(t))$. Next, we quantified the mean synchrony $R$ in each segment by $R = \langle r(t) \rangle = L^{-1}\sum_{t=1}^{L} r(t)$, where $L$ is the length of the data segment in samples and $r(t)$ is the Kuramoto order parameter: $r(t) = N^{-1}\left|\sum_{j=1}^{N} e^{i\theta_j(t)}\right|$, which was used as a time-dependent measure of phase synchrony. Here, $N$ is the number of channels in the data segment. The length of the segment in samples $L$ is the product of the time segment considered (0.4 s) and the sampling frequency (2000 Hz), that is $L = 800$.

## Acknowledgements

We thank the members of the Plenz lab for helpful discussion and John Hylton for help in spike sorting. This work was supported by the Intramural Research Program of the National Institute of Mental Health (NIMH) and Conselho Nacional de Desenvolvimento Científico e Tecnológico (CNPq).

# Additional information

### Funding

| Funder | Author |
|---|---|
| National Institute of Mental Health | Shan Yu<br>Tiago L Ribeiro<br>Christian Meisel<br>Samantha Chou<br>Andrew Mitz<br>Richard Saunders<br>Dietmar Plenz |
| Conselho Nacional de Desenvolvimento Científico e Tecnológico | Tiago L Ribeiro |

The funders had no role in study design, data collection and interpretation, or the decision to submit the work for publication.

### Author contributions

Shan Yu, Conceptualization, Data curation, Software, Formal analysis, Supervision, Validation, Investigation, Visualization, Methodology, Writing—original draft, Writing—review and editing; Tiago L Ribeiro, Software, Formal analysis, Investigation, Visualization, Methodology, Writing—original draft, Writing—review and editing; Christian Meisel, Software, Formal analysis, Visualization, Methodology,

Writing—review and editing; Samantha Chou, Data curation, Writing—review and editing; Andrew Mitz, Richard Saunders, Resources, Data curation, Methodology; Dietmar Plenz, Conceptualization, Resources, Software, Supervision, Funding acquisition, Validation, Investigation, Visualization, Methodology, Writing—original draft, Project administration, Writing—review and editing

### Author ORCIDs
Shan Yu (iD) http://orcid.org/0000-0002-9008-6658
Tiago L Ribeiro (iD) http://orcid.org/0000-0003-3195-9284
Andrew Mitz (iD) http://orcid.org/0000-0002-8045-0970
Dietmar Plenz (iD) http://orcid.org/0000-0002-0008-3657

### Ethics
Animal experimentation: All procedures followed the Institute of Laboratory Animal Research (part of the National Research Council of the National Academy of Sciences) guidelines and were approved by the NIMH Animal Care and Use Committee.(protocol #LSN-11).

### Decision letter and Author response
Decision letter https://doi.org/10.7554/eLife.27119.011
Author response https://doi.org/10.7554/eLife.27119.012

## Additional files

### Supplementary files
• Transparent reporting form
DOI: https://doi.org/10.7554/eLife.27119.010

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
