## [Decision Letter]

Thank you for submitting your article "Maintained avalanche dynamics during task-induced changes of neuronal activity in nonhuman primates" for consideration by *eLife*. Your article has been favorably evaluated by a Senior Editor and three reviewers, one of whom is a member of our Board of Reviewing Editors. The following individual involved in review of your submission has agreed to reveal his identity: Woodrow Shew (Reviewer #2). We have worked together to provide you with a consensus decision letter to help you prepare a revision.

This manuscript sets out to explore the hypothesis that task-related population dynamics in the cerebral cortex of behaving monkeys are organized according to a special type of scale-invariant spatiotemporal statistics called neuronal avalanches. This is an important hypothesis because previous work has shown that cortical states with neuronal avalanches are associated with functional advantages. However, until this paper, no one has carefully examined how and whether avalanche dynamics manifest during the execution of a well-defined behavioral task. The paper's primary, and most important finding confirms that task-related cortical activity is indeed organized as neuronal avalanches. Secondary to this, the authors have developed a new method of analyzing nonstationary data (as all task-related activity is) in the framework of neuronal avalanches.

Major Issues for Revision:

1) Issues related to the statistical robustness of the results

1a) The primary point of the paper is about the existence of neuronal avalanches during task-evoked cortical activity. However, most of the figures and work presented in the paper are focused on their new method of avalanche analysis using "adaptive binning". The authors claim that without adaptive binning, their measured avalanche size distributions deviate from power-law (subsection “Avalanche maintenance during self-initiated movements”, third paragraph and subsection “Transient rate changes during cue-triggered cognitive tasks”, last paragraph). These are some of the most important, but also weakest claims in the paper given that it is not clear they are supported by quantitative statistical measures. To a naive reader, the BASE and PRE distributions in Figure 2 appear to be extremely similar to each other. Without quantitative statistical support, the reader is left to suspect that the differences are not actually statistically significant. Perhaps they are slightly less similar to each other than those in Figure 2, but this seems to be a very subtle difference. Without some quantitative, statistically rigorous support for this claim, much of the following work seems poorly motivated. In other words, if there was no significant deviation from power-law to begin with, then adaptive binning would seem to be unnecessary.

1b) Are the avalanches shown in Figure 1 really invariant in timescale? Is trial averaging the electrodes obscuring a lot of variance in deflection size? And is light green line the overall average? "All electrodes" could also imply that should be the more variable set of traces showing individuals.

1c) The authors suggest that the maintenance of scale invariant avalanches could also maintain the functional advantages associated with neuronal avalanches. It would greatly improve their paper if they could directly test whether function is actually maintained. Perhaps one way to do this would be to follow up on the approach taken in Figure 5. For monkey B, they might compare some aspect of task performance (reaction time, success rate, etc.) for low rate vs. high rate trials. If function is not significantly different between these two groups of trials, it might strengthen the claim that function is maintained in spite of rate changes in avalanches. Or, if function is different for high rate vs. low rate trials, this might cast some doubt on the hypothesis that function is maintained throughout the changes in rate.

1d) In Figure 4, there is, indeed a shift toward increased rates and smaller changes in size compared to B however there are still statistical changes in avalanche size unlike Figure 2. How do you account for this?

2) Conceptual issues:

2a) Although the dataset of Yu et al. shows rather subtle (if any) deviations from power-law avalanche distributions, other datasets, like that of Shew et al. 2015, seem to show bigger deviations from power-laws at the onset of visual stimulation, for example. This raises the question: would adaptive binning rescue the power-law distribution for all levels of increased rate, or just for sufficiently small increases in rate? This is important given that a sufficiently strong stimulus can evoke a saturated response, i.e. all or nearly all electrodes exhibiting response in unison. Can such an extremely saturated response really be scale invariant? It has one scale – the system size. We are not suggesting that the monkeys in the experiments presented here are in this saturated regime. But, in principle, this could happen, particularly in primary sensory cortex. This seems to be an important limiting case, in which adaptive binning would be expected to fail. This should be mentioned as a limitation of the adaptive binning method. This case should also be discussed as an example in which sufficiently strong input could create cortical dynamics that are not expected to be scale invariant even if the underlying network is 'truly' critical.

This is not just a conceptual point about whether cortex always supports scale invariant dynamics; it is also likely to matter for function. For example, if we are subjected to a strong enough increase in sensory input, there will be a brief transient period during which the details of the stimulus are difficult to perceive, i.e. function is suboptimal for a brief transient period. For example, leaving darkness and entering a very bright space, it takes some time (~ few hundred ms) to adjust before you can see well. In other words, not only should we expect a deviation from neuronal avalanches during such an intense input, we should also expect degraded function. As suggested by Shew et al. 2015 Nat Phys and many others, it seems that adaptation (tuning of synapses) is necessary to overcome the poor sensory processing that comes with a strongly saturated response.

2b) It is confusing when the authors reference the hard-core physics notion that criticality cannot really exist in a driven system. If that's the case, it raises the question: why do any of this analysis? Perhaps, it would be helpful to mention that the hard-core physics notion of true criticality may not be relevant given that every real neural system has some external input. The functional advantages seem to be robust to some amount of external input.

2c) It has previously been suggested that networks depart from criticality during task-evoked activity. However, given the mathematical advantages of critical dynamics, it would be curious if networks deviated precisely at the time when they should be processing task-relevant information. This paper presents a compelling explanation, which is that increases in average activity correspond to a higher avalanche frequency, and using inappropriate time bins effectively lumps multiple cascades together making the network appear to exhibit supercritical dynamics. Indeed, with shorter time bins during task-evoked increases in activity the number of large avalanches is reduced. Given all of this, it's not clear why this shorter time bin shouldn't be an appropriate size for the entire data set. In other words, if the argument is that it's only the rate and not the size of the avalanches that change, then shouldn't this smaller time bin still capture all avalanches and just show that there are fewer of them in the baseline and post-event epochs? Or is it simply more generalizable to recommend estimating the bin size based on local statistics rather than searching the data set for the optimal bin size?

2d) Related to this, adaptive binning resulted in longer time bins in the post-event epoch, but this did not normalize the avalanche statistics as it did in the pre-event epoch. At this time there were still fewer and smaller avalanches than baseline (Figure 2). How is this explained? Is this evidence against criticality?

3) Issues concerning interpretation

3a) Reinterpreting large avalanches as multi-avalanches is a reasonable and interesting thing to try, but was guaranteed to shift size into rate as shown. Therefore, the main result is really the tighter power law curves. However, we would like to see a more careful consideration of alternative hypotheses?

3b) Somewhere in the Introduction or Discussion, it would be beneficial to more clearly explain the difference between a system with 'true' criticality but without scale invariant dynamics versus a truly non-critical system. Both would lack scale invariant dynamics. The hypothesis proposed by the authors is that their experiments are a case of the former (true criticality without power-law avalanches). However, the alternative, competing hypothesis is also quite plausible. That is, there are many well-known ways that interactions among neurons can change (synaptic depression, facilitation, neuromodulators, etc.) and therefore true deviations from criticality might result. I think the paper would be stronger if this alternative was explained.

3c) From the perspective of a researcher, the adaptive binning method seems to be a useful tool for learning something about the state of the cortex in a way that is less dependent on activity rate nonstationarities. However, adaptive binning may be irrelevant from the perspective of a downstream neuron that receives input from a region with changing rate. Unless the downstream neuron has some way of adapting its own synaptic integration timescales, it seems like it would see input that deviates from scale-free avalanches.

3d) The authors suggest that the functional benefits associated with neuronal avalanches might be preserved, but it seems that, functionally, real neurons do not have the ability to do the adaptive binning they would need to get the functional benefits of neuronal avalanches. Moreover, rate coding is prominent in many neural systems. If neurons do implement some kind of adaptive binning, it seems they would be ignoring rate coded information. Please discuss this in the Discussion section.

---

## [Author Response]

Major Issues for Revision:1) Issues related to the statistical robustness of the results1a) The primary point of the paper is about the existence of neuronal avalanches during task-evoked cortical activity. However, most of the figures and work presented in the paper are focused on their new method of avalanche analysis using "adaptive binning". The authors claim that without adaptive binning, their measured avalanche size distributions deviate from power-law (subsection “Avalanche maintenance during self-initiated movements”, third paragraph and subsection “Transient rate changes during cue-triggered cognitive tasks”, last paragraph). These are some of the most important, but also weakest claims in the paper given that it is not clear they are supported by quantitative statistical measures. To a naive reader, the BASE and PRE distributions in Figure 2 appear to be extremely similar to each other. Without quantitative statistical support, the reader is left to suspect that the differences are not actually statistically significant. Perhaps they are slightly less similar to each other than those in Figure 2, but this seems to be a very subtle difference. Without some quantitative, statistically rigorous support for this claim, much of the following work seems poorly motivated. In other words, if there was no significant deviation from power-law to begin with, then adaptive binning would seem to be unnecessary.

We thank the reviewers for these comments. In fact, significance was demonstrated in our original submission using a boot-strap method. This provided shaded confidence intervals, which unfortunately, made differences visually less distinctive. In response, we now provide separate figures for each monkey that detail differences in size distributions and we added additional quantifications of differences (new Figure 3 and Figure 6). These figures now show each size distribution normalized by the expected power law. Thus, the expected power law translates into a horizontal curve, which allows power-law deviations to be visualized clearly. The area under each normalized distribution now quantifies how much LATE and PRE distributions are biased towards larger avalanches. Finally, we present the actual distribution of differences in the mean avalanche size of LATE/PRE and BASE, for each trial, using fixed and adaptive binning. Comparing these distributions demonstrate the significant effect in adaptive binning in recovering a power law. As a minor point, we now present the distributions of the monkey performing the cognitive task first.

1b) Are the avalanches shown in Figure 1 really invariant in timescale? Is trial averaging the electrodes obscuring a lot of variance in deflection size? And is light green line the overall average? "All electrodes" could also imply that should be the more variable set of traces showing individuals.

We apologize for the confusion. Figure 1 (now Figure 4) displayed the average, touch-pad aligned LFP waveform for each single electrode superimposed and the corresponding average (center line). These data support the extraction of negative LFP threshold crossings (nLFPs) to identify nLFP rate changes that identify BASE, PRE and POST periods. We have reworded the figure legends accordingly to avoid this confusion.

1c) The authors suggest that the maintenance of scale invariant avalanches could also maintain the functional advantages associated with neuronal avalanches. It would greatly improve their paper if they could directly test whether function is actually maintained. Perhaps one way to do this would be to follow up on the approach taken in Figure 5. For monkey B, they might compare some aspect of task performance (reaction time, success rate, etc.) for low rate vs. high rate trials. If function is not significantly different between these two groups of trials, it might strengthen the claim that function is maintained in spite of rate changes in avalanches. Or, if function is different for high rate vs. low rate trials, this might cast some doubt on the hypothesis that function is maintained throughout the changes in rate.

We thank the reviewer for this important question given the performance task we analyzed. In fact, we test both trial outcome and reaction time. Regarding trial outcome, the monkey’s success was > 95% leaving us with too few false trials to establish statistical significance. We then extracted reaction time as the period from the visual cue being switched off to reaching the feeder. There was no correlation between the number of nLFPs on the array and reaction time. Similarly, no correlation was found for avalanche size and rate using adaptive or fixed binning. These results, though negative, demonstrate that adaptive binning does introduce new correlations and have now been added to the manuscript (subsection “Avalanche maintenance during a cue-triggered cognitive task”, last paragraph).

We have also added new paragraphs in the Discussion (seventh and eighth paragraphs) which elaborates on the issue that the dynamical state (as identified by power law statistics) is maintained in the monkeys independent from task-induced activity rate changes. It’s important that we emphasize that our message is that the dynamical state is maintained regardless of rate changes. That doesn’t necessarily imply that the performance doesn’t change with rate: the fact that some trials have decreased rate could be linked to decreased input, which could be, for instance, a reflection of lack of attention, which would lead to worse performance, regardless of the dynamical state of the network in pre-frontal cortex. The fact that this network is critical means it can do the best possible processing given the input it received. The only true way of testing the link between performance and dynamical state is by disturbing the dynamical state of the network we’re investigating.

1d) In Figure 4, there is, indeed a shift toward increased rates and smaller changes in size compared to B however there are still statistical changes in avalanche size unlike Figure 2. How do you account for this?

In our new Figure 3 we now show the statistics of significantly reduced bias towards larger avalanches for adaptive binning. This figure also quantifies the residual error this referee pointed out. We now discuss potential sources of this residual error in the manuscript. We believe one major source originates from the necessity to parse the behavioral continuous time course into three episodes from average trial activity for which adaptive bins are calculated. This error is inherent to any approach of estimating rate from a non-zero time period. A second error arises when there are too few events, for e.g. during the POST epoch, when activity is bounded by a zero-activity floor. These limitations are now discussed in a separate paragraph in the Discussion (seventh paragraph).

2) Conceptual issues:2a) Although the dataset of Yu et al. shows rather subtle (if any) deviations from power-law avalanche distributions, other datasets, like that of Shew et al. 2015, seem to show bigger deviations from power-laws at the onset of visual stimulation, for example. This raises the question: would adaptive binning rescue the power-law distribution for all levels of increased rate, or just for sufficiently small increases in rate? This is important given that a sufficiently strong stimulus can evoke a saturated response, i.e. all or nearly all electrodes exhibiting response in unison. Can such an extremely saturated response really be scale invariant? It has one scale – the system size. We are not suggesting that the monkeys in the experiments presented here are in this saturated regime. But, in principle, this could happen, particularly in primary sensory cortex. This seems to be an important limiting case, in which adaptive binning would be expected to fail. This should be mentioned as a limitation of the adaptive binning method. This case should also be discussed as an example in which sufficiently strong input could create cortical dynamics that are not expected to be scale invariant even if the underlying network is 'truly' critical.This is not just a conceptual point about whether cortex always supports scale invariant dynamics; it is also likely to matter for function. For example, if we are subjected to a strong enough increase in sensory input, there will be a brief transient period during which the details of the stimulus are difficult to perceive, i.e. function is suboptimal for a brief transient period. For example, leaving darkness and entering a very bright space, it takes some time (~ few hundred ms) to adjust before you can see well. In other words, not only should we expect a deviation from neuronal avalanches during such an intense input, we should also expect degraded function. As suggested by Shew et al. 2015 Nat Phys and many others, it seems that adaptation (tuning of synapses) is necessary to overcome the poor sensory processing that comes with a strongly saturated response.

We agree with the reviewer and have included a discussion of saturated responses to the method in the appropriate section (Discussion, seventh paragraph).

2b) It is confusing when the authors reference the hard-core physics notion that criticality cannot really exist in a driven system. If that's the case, it raises the question: why do any of this analysis? Perhaps, it would be helpful to mention that the hard-core physics notion of true criticality may not be relevant given that every real neural system has some external input. The functional advantages seem to be robust to some amount of external input.

The reviewer is correct and we have modified the main text according to his/her suggestions (Discussion, eleventh paragraph).

2c) It has previously been suggested that networks depart from criticality during task-evoked activity. However, given the mathematical advantages of critical dynamics, it would be curious if networks deviated precisely at the time when they should be processing task-relevant information. This paper presents a compelling explanation, which is that increases in average activity correspond to a higher avalanche frequency, and using inappropriate time bins effectively lumps multiple cascades together making the network appear to exhibit supercritical dynamics. Indeed, with shorter time bins during task-evoked increases in activity the number of large avalanches is reduced. Given all of this, it's not clear why this shorter time bin shouldn't be an appropriate size for the entire data set. In other words, if the argument is that it's only the rate and not the size of the avalanches that change, then shouldn't this smaller time bin still capture all avalanches and just show that there are fewer of them in the baseline and post-event epochs? Or is it simply more generalizable to recommend estimating the bin size based on local statistics rather than searching the data set for the optimal bin size?

The issue with using a single bin to resolve all data is the non-stationarities, and the explanation is in (old) Figure 5. Using a small bin for all the data would be equivalent of setting the bin at the orange level, let’s assume. That would result in power laws with different exponents which when combined would lead to bumps as observed in the grey curve. It’s important to notice that the bin is calculated for every trial, and not just one for each epoch. We added this explanation to the main text (subsection “Impact of ‘adaptive binning’ in capturing rate change vs. change in dynamics”, first paragraph).

2d) Related to this, adaptive binning resulted in longer time bins in the post-event epoch, but this did not normalize the avalanche statistics as it did in the pre-event epoch. At this time there were still fewer and smaller avalanches than baseline (Figure 2). How is this explained? Is this evidence against criticality?

We agree with this reviewer and have now added an extensive paragraph in the Discussion about the limitation of adaptive binning. In general, saturation, when responses cannot be discriminated anymore and very low-activity, when even long time bins can’t concatenate events into clusters for the recording duration given, provide upper and lower limits respectively (see Discussion, seventh paragraph).

3) Issues concerning interpretation3a) Reinterpreting large avalanches as multi-avalanches is a reasonable and interesting thing to try, but was guaranteed to shift size into rate as shown. Therefore, the main result is really the tighter power law curves. However, we would like to see a more careful consideration of alternative hypotheses?

In fact, our simulations considered three alternative (non-critical) hypotheses (Figure 7): supercritical, added noise, and faster propagation (not shown in the manuscript due to non-physiologically high propagation rates to obtain observed rate changes). All of them failed to reproduce the observed experimental results. We also provided two additional methods (Figure 8; end of Results section): adaptive thresholding and synchronization measure, which are completely separated and independent of temporal binning. All are in line with the results observed for adaptive binning supporting that the increased drive is the most likely explanation for the data. We believe that the revision of the manuscript and additional analysis provided have increased the transparency of our work such that these alternative considerations stand out more.

3b) Somewhere in the Introduction or Discussion, it would be beneficial to more clearly explain the difference between a system with 'true' criticality but without scale invariant dynamics versus a truly non-critical system. Both would lack scale invariant dynamics. The hypothesis proposed by the authors is that their experiments are a case of the former (true criticality without power-law avalanches). However, the alternative, competing hypothesis is also quite plausible. That is, there are many well-known ways that interactions among neurons can change (synaptic depression, facilitation, neuromodulators, etc.) and therefore true deviations from criticality might result. I think the paper would be stronger if this alternative was explained.

We agree with the reviewer that is fundamentally important to discuss the alternative that a true deviation from criticality is the reason for the deviation towards larger avalanches in the data. I believe we addressed this point by providing 8 references in the Introduction (last paragraph) before we present our hypothesis as alternative view. We used the Results section to detail our simulations in which true deviations from criticality are explored further. Accordingly, we show in these simulations that in a truly supercritical system, our approach of adaptive binning leads to even larger deviations from power law distributions, which is in absolute contrast with the experimental data. See point 3a above. We believe our current layout of addressing non-critical dynamics and our suggested approach works well given the space requirements and flow of the manuscript.

3c) From the perspective of a researcher, the adaptive binning method seems to be a useful tool for learning something about the state of the cortex in a way that is less dependent on activity rate nonstationarities. However, adaptive binning may be irrelevant from the perspective of a downstream neuron that receives input from a region with changing rate. Unless the downstream neuron has some way of adapting its own synaptic integration timescales, it seems like it would see input that deviates from scale-free avalanches.

We thank the reviewer for this insightful comment. We now reference work in which neurons have been suggested to reduce their integration time constant as they receive more input, which essentially mimics adaptive binning as a physiological process. We have added this point to the Discussion section (eighth paragraph).

3d) The authors suggest that the functional benefits associated with neuronal avalanches might be preserved, but it seems that, functionally, real neurons do not have the ability to do the adaptive binning they would need to get the functional benefits of neuronal avalanches. Moreover, rate coding is prominent in many neural systems. If neurons do implement some kind of adaptive binning, it seems they would be ignoring rate coded information. Please discuss this in the Discussion section.

We thank the reviewer for this comment. We added that to the Discussion section (see point 3c above).